# Skin-Based Vaccination: A Systematic Mapping Review of the Types of Vaccines and Methods Used and Immunity and Protection Elicited in Pigs

**DOI:** 10.3390/vaccines11020450

**Published:** 2023-02-16

**Authors:** Inés Có-Rives, Ann Ying-An Chen, Anne C. Moore

**Affiliations:** School of Biochemistry and Cell Biology, University College Cork, T12 XF62 Cork, Ireland

**Keywords:** vaccine, pig, skin, intradermal, transdermal, transcutaneous, epidermal, epicutaneous, percutaneous, needle-free

## Abstract

The advantages of skin-based vaccination include induction of strong immunity, dose-sparing, and ease of administration. Several technologies for skin-based immunisation in humans are being developed to maximise these key advantages. This route is more conventionally used in veterinary medicine. Skin-based vaccination of pigs is of high relevance due to their anatomical, physiological, and immunological similarities to humans, as well as being a source of zoonotic diseases and their livestock value. We conducted a systematic mapping review, focusing on vaccine-induced immunity and safety after the skin immunisation of pigs. Veterinary vaccines, specifically anti-viral vaccines, predominated in the literature. The safe and potent skin administration to pigs of adjuvanted vaccines, particularly emulsions, are frequently documented. Multiple methods of skin immunisation exist; however, there is a lack of consistent terminology and accurate descriptions of the route and device. Antibody responses, compared to other immune correlates, are most frequently reported. There is a lack of research on the underlying mechanisms of action and breadth of responses. Nevertheless, encouraging results, both in safety and immunogenicity, were observed after skin vaccination that were often comparable to or superior the intramuscular route. Further research in this area will underlie the development of enhanced skin vaccine strategies for pigs, other animals and humans.

## 1. Introduction

Skin-based immunisation is defined as the administration of a vaccine into, onto or through the skin. This route of administration is not new; the inoculation into the skin of variola virus, termed variolation, popularly dates back at least to the time of the Ottoman Empire in the seventeenth century, with other reports from China in the eleventh century. Variolation consisted of rubbing contents of virus-laden pustules onto scratched skin in an attempt to protect against fatal smallpox [1]. This ancient technique was further developed by Edward Jenner’s use of cowpox instead of smallpox pustules, which was also administered to superficial incisions on the skin. Scarification, consisting of scraping the skin to break the outer layers with a lancet or, in modern times, a bifurcated needle, was subsequently adopted as the vaccination delivery technique [2]. The smallpox vaccine, given by the skin route, led to the first and only, eradication of a human disease in 1980 [3].

The skin harbours cells from both the innate and the adaptive immune system [4]. This has been suggested as a reason why stronger systemic immune responses can be elicited by skin-based vaccination. Potent responses may permit vaccine dose-sparing, already proven for influenza, rabies and hepatitis B vaccines [5,6]. This dose-sparing advantage of skin-based immunization could be key to overcome vaccine shortages. In addition, new delivery technologies, such as microneedle patches, are being developed, primarily for human use. Some of these should ease the administration process, reducing the need for highly trained personnel [7,8,9,10]. It is, therefore, timely to review the state of the art with respect to skin-based vaccination. Within this field, the pig is a suitable large animal model due to their anatomical and immunological similarity to humans [11,12,13,14]. Their central role in the One Health approach as food industry animals [15] and source of zoonotic diseases [16] also situates them as a target species for some vaccines [17,18]. The aim of this review is to evaluate the field of skin-based vaccination in the pig, focusing on vaccine-induced immune responses and safety. 

### 1.1. Skin-Based Vaccination

The skin is an attractive vaccine administration site due to its physiological and immunological characteristics. It is the biggest and outermost organ in the human body and consists of four distinct layers: the outermost barrier; the stratum corneum, then the epidermis, dermis, and hypodermis (Figure 1a). The stratum corneum is a lipid barrier to the surrounding environment that must be traversed to gain access to deeper skin layers containing immune cells [4]. The abundant presence of components of the immune system in the skin has led to the concept that targeting vaccines to this immune-rich organ could have benefits with respect to vaccine potency and the magnitude and quality of the response. Safety and strong immunogenicity of vaccines delivered by the skin route has been reported in pre-clinical [19,20,21,22,23] and clinical trials of human-targeting vaccines [5,24,25,26]. The presence of different subsets of antigen presenting cells (APCs) in the different layers of the skin is of special interest, suggesting a potentially distinct response depending on the cells targeted. Langerhans cells and dermal dendritic cells are mainly found in the epidermis and dermis, respectively [4,27,28,29,30] (Figure 1b), and they have key roles in driving humoral and cellular immune responses [31,32,33,34,35]. Langerhans cells have dichotomous roles, balancing between immunity and tolerance [36]. In addition, they have been shown to be able to prime CD8^+^ T cells and CD4^+^ Th2 cells [33,35,37]. In contrast, dermal dendritic cells have been associated with inducing CD4^+^ Th1 cells [37] and having a faster migration to differential locations in the lymph nodes [38]. Lymphocytes can also be found in the skin. Resident T lymphocytes, mainly CD4^+^ T cells, are found in high quantities in human skin, under steady state conditions [39]. Most lymphocytes localise to the dermis, with the exception of a low percentage of CD8^+^ T lymphocytes found in mouse and human epidermis [39,40,41] (Figure 1b). In humans, T cells express diverse TCR repertoire and belong preferentially to the alpha/beta population, with very few belonging to the gamma/delta subset [39,42,43]. In contrast, in mice, gamma/delta T cells constitute the majority of T cells in epidermis and dermis [42,44,45]. T cells in human skin predominantly show a memory effector phenotype (CD45RO^+^, CD62L−, CCR7−), hence they are ready for an immediate protective effector response when re-encountering antigens [39,46]. The presence of a memory [40] and plasma [47] resident B cells phenotype in the skin has also been recently demonstrated. Thus, rapid action by these cells is expected upon antigen stimulation. Furthermore, skin immunisation has been proposed to provide a greater magnitude of response by inducing mucosal immune responses in mice and pigs capable of fighting pathogens at their entry site [19,23,48,49,50,51].

Potentially stronger immune responses at the skin level could permit the use of reduced antigen doses in a vaccine without compromising efficacy and/or for a broader and stronger response to be induced by a full dose of the vaccine, which could potentially lead to single dose vaccines. The potential of skin-based vaccination to elicit broader immune responses, which has been demonstrated in mice [52,53], would contribute to making currently licensed vaccines [52] and developing vaccine platforms [53] more universal. Dose-sparing has been proven in clinical trials, with immunogenicity being at least non-inferior to that of intramuscular vaccination for influenza, rabies, and hepatitis B vaccines [5,6]. Thus, the smaller dose required for the skin-based route of immunisation could facilitate wider distribution of vaccines. This is crucial for emergency response during epidemics and pandemics in addition to enhancing routine immunisation programmes [54]. Furthermore, this strategy would address ever-rising vaccine shortages [55]. However, the dose-sparing effect is not observed for all skin-delivered vaccines. For example, the use of fractional dose inactivated polio vaccine (IPV) administered intradermally demonstrated contradictory results in clinical trials [5]. While some trials have observed comparable seroconversion rates between fractional and full doses [56,57], others have highlighted the inferiority of the reduced dose in eliciting immune response, especially in very young infants [58,59,60]. In addition, scarce availability of studies for some skin-delivered vaccines make it difficult to extract precise conclusions, despite the promising results observed [5]. Thus, the dose-sparing effect is a possibility worth further evaluation in clinical trials, considering potential differences between vaccines, modes of injection and vaccination subjects, which will need to be adjusted.

Ease of administration is another key factor needed for successful immunisation campaigns. Skin vaccination using scarifiers, needles and syringes using the intradermal Mantoux technique was conventional practice for the smallpox vaccine, but presents limitations regarding the reproducibility of administered quantities and precision at the site of injection [61,62,63]. New technologies aim to allow precise and easy-to-perform administration, thereby reducing or eliminating the need for highly trained personnel. This will contribute to increased compliance and vaccine uptake. They consist of needle-free or minimally invasive administration techniques at the epidermis and dermis levels, including small needles, jet injectors, gene guns, and microneedle array patches (MAPs) [7,9,64,65]. These techniques have also proven to be immunogenic in animal models [23,52,53,66,67,68], humans [24,69,70,71,72,73,74,75], and veterinary medicine [76,77]. The use of jet injectors in veterinary medicine has increased in recent decades, with some being available for the routine immunisation of pigs [78,79]. 

The strong immunogenic potential, the potential for dose-sparing, and development of easy-to-use delivery technologies have led to an increased focus on the use of the skin route for a wide range of vaccines. However, skin immunisation still has challenges to overcome, including inferiority in immunogenicity observed for some vaccines when compared to the intramuscular route and adaptation of vaccine formulation to new delivery methods. 

### 1.2. Pigs as a Model for Skin-Based Vaccination

The most used animal model in biomedical research is the mouse. However, they present limitations, particularly with respect to lack of physiological and immunological similarity to humans [45]. Focusing on skin-based immunisation, swine are a highly relevant model for research on the immune system and the skin [80,81,82,83,84,85]. Pigs are used in a wide range of research fields and they share physiological, anatomical and immunological similarities with humans [86]. The genes of pigs’ immune system show preservation of orthology with human genes, 13 times higher than between human and mice, being especially representative in members of the chemokine and cytokine families [13,14]. There is a higher percentage of similarity between protein sequences and structures, with predicted responses for some chemokines, pattern recognition receptors and T lymphocytes being better preserved between pigs and humans compared with mice and humans [13]. Moreover, organs from the immune system, such as the thymus, present similarities in structure and development although some differences, for example, the presence of inverted structure of mucosa-associated lymph nodes in the pig, are observed [87]. 

The skin of pigs is anatomically very similar to that of humans, with similar thickness and composition of epidermis and dermis and both presenting age-dependant differences [11,12,88]. Chemical and structural closeness has also been observed [89], with similar permeability [11] and mechanical properties [90], which are very useful characteristics for vaccine research. There is also a resemblance between the two species at the cellular level. Studies have identified pig dendritic cell subsets that are phenotypically and functionally similar to their human counterparts, fitting the classification of Langerhans cells and dermal DCs that are found in similar locations in both species [91,92,93]. Regarding the adaptive immune system, markers for both B and T cells and migration patterns are similar to those used in humans [94,95]. Porcine B cells have been observed to respond to TLR ligands in a comparable way to humans, leading to their differentiation [95]. Pigs, like humans, also possess alpha/beta T cells, which are classified according to their CD4 and CD8 markers. However, some differences do exist. Anatomically, pigs have a decreased number and size of elastic fibres and a higher number of blood vessels compared with humans. Pigs, unlike humans, have some granulose cells near the blood vessels and present apocrine sweat glands instead of eccrine glands [11]. Immunologically, the proportion of alpha/beta and gamma/delta T cells in blood is reversed, especially in young pigs, with a gradual increase in the proportion of alpha/beta T cells observed with age [94]. Aside from their role in wound healing and homeostasis [44], gamma/delta T cells have been suggested to have an active role in the early phases of immune response to infection and/or vaccination [96]. Circulating CD4^+^ CD8^+^ double positive T cells can be found in a higher percentage in healthy pigs compared with humans, and have been associated with memory and effector functions [97,98]. Understanding the anatomical and immunological similarities and differences to humans renders pigs as relevant models for skin-based vaccination studies.

### 1.3. One Health Perspectives

One Health is defined by the One Health High Level Expert Panel as “an integrated, unifying approach that aims to sustainably balance and optimize the health of people, animals, and ecosystems. It recognizes the health of humans, domestic and wild animals, plants, and the wider environment (including ecosystems) are closely linked and inter-dependent” [99]. Pigs are a source of zoonotic diseases [16], which makes them a crucial part of the One Health approach. As an example, the 2009 pandemic influenza virus derived from pigs when human, avian and pig influenza viruses reassorted, leading to this novel, pandemic H1N1 virus strain [100,101]. The presence of human and avian influenza receptors in pigs leads to the perfect environment for genetic reassortment of viruses that can then potentially infect humans and lead to epidemics and pandemics [102]. This highlights the centrality of pigs in relation to infectious diseases, such as influenza and Nipah viruses, that are of high concern to human spill over events of pandemic potential. This further validates their use as models for human diseases [82,103] and vaccine development [17,18], particularly for pathogens shared with humans [82]. 

Pigs are also important livestock animals, with pork being the most consumed animal worldwide (36%) according to the UN-FAO. The European Union is one of the biggest exporters and the second biggest producer of pork after China [15]. High quality pig health care, including preventive and therapeutic treatments such as vaccination, are required to ensure food security. Vaccine development in pigs is a growing field, with respect to vaccines specific for pigs and also their use as a large animal model for human vaccines. Multiple vaccines for pigs are licensed (Table 1). Six intradermally delivered vaccines have been licensed at least in one member state of the European Union. The devices used for intradermal administration of these vaccines are jet injectors. 

As the interest in skin-based vaccination in human medicine is growing, it is timely to review developments in relation to skin-based vaccination in pigs. The aim of this review is to perform a comprehensive evaluation and to map the literature relating to skin-based vaccination in pigs. We focus on reviewing vaccine-induced humoral and cellular immune responses assessed at a systemic and mucosal level, as well as safety assessments. We quantitatively analyse the relative focus on different pathogens, the type of vaccine platforms and adjuvants used as well as the vaccination route and device. The current state of knowledge, gaps and future directions in this area are discussed. 

**Table 1 vaccines-11-00450-t001:** Pig vaccines against infectious pathogens centrally licensed by EMA [104] or by at least one EU-member state [105].

Target Pathogen	Vaccine Name	Type of Vaccine	Route of Administration	Marketing Authorisation Holder
*Lawsonia intracellularis*	Porcilis Lawsonia ID ^a^	Inactivated	ID	Intervet Ireland Limited
*Mycoplasma hyopneumoniae*	M Hyo ID ONCE ^a^	Inactivated	ID	Intervet Ireland Limited
*Mycoplasma hyopneumoniae* + PCV2 ^b^	Mhyosphere PCV ID	Inactivated (Mhyo)/recombinant subunit (PCV2)	ID	HIPRA
PCV2	Porcilis PCV ID	Subunit	ID	Intervet International BV
PRRSV ^b^	Porcilis© PRRS	Live attenuated	ID	Intervet International BV
PRRSV	UNISTRAIN^®^ PRRS ^a^	Live attenuated	ID	HIPRA
*Actinobacillus pleuropneumoniae*	Coglapix	Inactivated	IM	Ceva-Phylaxia Veterinary Biologicals Co. Ltd.
*Bordetella bronchiseptica + Pasteurella multocida*	Rhiniseng	Inactivated (Bb)/subunit (Pm)	IM	HIPRA
*Bordetella bronchiseptica + Pasteurella multocida*	Porcilis AR-T DF	Inactivated (Bb)/subunit (Pm)	IM	Intervet International BV
*Clostridium difficile and perfringens*	Suiseng Diff/A	Subunit (toxoid)	IM	HIPRA
*Clostridium perfringens + Escherichia coli*	Enteroporc COLI AC	Subunit (toxoids + fimbriae)	IM	Ceva Santé Animal
*Clostridium perfringens + Escherichia coli*	Porcilis ColiClos	Subunit (toxoids + fimbriae)	IM	Intervet International BV
CSFV ^b^	Suvaxyn CSF Marker	Viral vector	IM	Zoetis
*Escherichia coli*	VEPURED	Subunit	IM	HIPRA
*Escherichia coli*	Enteroporc COLI	Subunit (fimbriae)	IM	Ceva Santé Animal
*Escherichia coli*	Ecoporc SHIGA	Subunit	IM	Ceva Santé Animal
*Escherichia coli*	Porcilis Porcoli Diluvac Forte	Subunit	IM	Intervet International BV
*Escherichia coli*	Neocolipor	Inactivated	IM	Boehringer Ingelheim Vetmedica GmbH
*Escherichia coli*	Coliprotec F4/F18	live	Oral	Elanco GmbH
*Erysipelothrix rhusiopathiae*	Eryseng	Inactivated	IM	HIPRA
FMDV ^b^	AFTOVAXPUR DOE	Inactivated	IM	Boehringer Ingelheim Vetmedica GmbH
Influenza virus	Respiporc FLUpan H1N1	Inactivated	IM	Ceva Santé Animale
Influenza virus	Respiporc Flu3	Inactivated	IM	Ceva Santé Animal
*Lawsonia intracellularis*	Porcilis Lawsonia ^a^	Inactivated	IM	Intervet Ireland Limited
*Mycoplasma hyopneumoniae*	Suvaxyn M Hyo ^a^	Inactivated	IM	Zoetis
*Mycoplasma hyopneumoniae*	Suvaxyn MH-One ^a^	Inactivated	IM	Zoetis
*Mycoplasma hyopneumoniae* + PCV2	CircoMax Myco	Inactivated (Mhyo)/inactivated recombinant chimeric (PCV2)	IM	Zoetis
*Mycoplasma hyopneumoniae* + PCV2	Suvaxyn Circo+MH RTU	Inactivated (Mhyo)/inactivated recombinant chimeric (PCV2)	IM	Zoetis
*Mycoplasma hyopneumoniae* + PCV2	Porcilis PCV M Hyo	Inactivated (Mhyo)/subunit (PCV2)	IM	Intervet International BV
Porcine parvovirus	Porcilis© Parvo ^a^	Inactivates	IM	Intervet Ireland Limited
PCV2	Circovac	Inactivated	IM	CEVA-PHYLAXIA
PCV2	Ingelvac CircoFLEX	Subunit	IM	Boehringer Ingelheim Vetmedica GmbH
PCV2	Porcilis PCV	Subunit	IM	Intervet International BV
PCV2	Suvaxyn Circo	inactivated recombinant chimeric	IM	Zoetis
PCV2	CircoMax	Inactivated recombinant chimeric	IM	Zoetis
Porcine parvovirus	ReproCyc ParvoFLEX	Subunit	IM	Boehringer Ingelheim Vetmedica GmbH
Porcine Parvovirus + *Erysipelothrix rhusiopathiae*	Eryseng Parvo	Inactivated	IM	HIPRA
Porcine Parvovirus + *Erysipelothrix rhusiopathiae*	BIOSUIS ParvoEry ^a^	Inactivated	IM	Bioveta
PRRSV	Suvaxyn PRRS MLV	Live attenuated	IM	Zoetis
PRRSV	UNISTRAIN^®^ PRRS ^a^	Live attenuated	IM	HIPRA
PRRSV	Ingelvac PRRSFLEX EU ^a^	Live attenuated	IM	Boehringer Ingelheim
PRRSV	Porcilis© PRRS	Live attenuated	IM	Intervet International BV
PRRSV	ReproCyc PRRS EU ^a^	Live attenuated	IM	Boehringer Ingelheim
PRV ^b^	Suvaxyn Aujeszky 783 + O/W	Live attenuated	IM	Zoetis
*Salmonella enterica*	BIOSUIS Salm ^a^	Inactivated	IM	Bioveta

^a^: Not central Marketing Authorisation. ^b^: Abbreviations PCV2; Porcine Circovirus type 2. CSFV; Classical Swine Fever Virus. FMDV; Food-and-Mouth Disease Virus. PRRSV; Porcine Reproductive and Respiratory Syndrome virus. PRV; Pseudorabies Virus.

## 2. Methodology

A PubMed search was conducted to identify published peer-reviewed literature that used skin routes of administration. The specific search was as follows: (((pigs OR porcine OR swine) AND (vaccin*)) AND (intradermal OR transdermal OR transcutaneous OR epidermal OR epicutaneous OR percutaneous OR needle-free OR skin OR scarification)) NOT (guinea). The inclusion criteria were (i) peer-reviewed papers; (ii) must be in the English language; (iii) contain primary data; (iv) feature pigs as animal model; (v) contain in vivo work; (vi) include a vaccine; (vii) inoculation to be performed through the skin route and (viii) evaluation of the safety and/or efficacy of vaccine. Efficacy includes protection or immune response that is a correlate of protection, elicited by the vaccine.

Each paper was analysed and classified according to the following parameters: immunisation regimens and vaccines used, including the administration route and device, target pathogen, type of vaccine platform, type of adjuvant and breed of pig, that could modulate the immunological endpoints. Where papers fit into more than one category in any variable, the paper was classified in all appropriate categories. 

The search returned with 389 articles published through 8 September 2022. Screening of the papers led to the exclusion of 272 papers which did not comply with at least one of the inclusion criteria set (Figure 2). A total of 117 papers were included in this review. The first author reviewed, screened and characterised all articles. The two other authors reviewed the categorisation and reviewed articles that required re-examination. The authors examined articles independently. No automation tools were used in the process. Frequencies of multiple variables within the literature selected were analysed using the ggplot2 package from RStudio.

## 3. Results and Discussion

### 3.1. Type of Pigs

There are 10 species of living pigs within the Sus genus including domestic pigs and their ancestors, wild boars [106]. Miniature pigs, micro-pigs and domestic pigs are all classified as the same species [107]. We analysed the frequency of use of each species and breed in the skin vaccination field and determined their proportionate use. Articles tend to use the term “commercial domesticated pigs”; this contains many breeds. Among the included literature, 35 papers use Yorkshire, also known as Large White, and Landrace, or crossbreeds of the same. In a few studies, minipigs were used as they are regarded as a suitable model when it comes to vaccination through the skin [83]. It is noted that almost half of the identified papers (n = 57), do not name the pig breed, often referring to it only as commercial breed [49,51,76,108,109,110,111,112,113,114,115,116,117,118,119,120,121,122,123,124,125,126,127,128,129,130,131,132,133,134,135,136,137,138,139,140,141,142,143,144,145,146,147,148,149,150,151,152,153,154,155,156,157,158,159,160,161]. All mentioned breeds are from domestic pigs. 

### 3.2. Vaccine Targets for Skin-Based Immunisation in Pigs

We next determined the range and frequencies of the pathogens targeted by vaccines using the skin route of administration. The pathogens were first defined as viruses, bacteria, and parasites and a fourth category, labelled “other”, which includes studies using vaccines consisting of model antigens (namely ovalbumin, OVA and keyhole limpet haemocyanin, KLH). Almost 80% of skin-delivered vaccines target viruses (Figure 3). The remainder focused on bacteria (16.5%), parasites (2.5%) and other targets (5%).

Within viruses, it is evident that vaccines targeted to PRRSV and influenza viruses predominate (Figure 3). This reflects the magnitude of the PRRSV problem and the importance of pigs as a zoonotic source in influenza vaccine development efforts in addition to vaccines for swine influenza viruses. 

Porcine Reproductive and Respiratory Syndrome virus (PRRSV) is an arterivirus, which are enveloped, positive strand RNA virus. It is one of the leading diseases affecting farm pigs and a cause of major economic losses, with the EU estimating an annual cost of €1.5 billion and the US reporting an estimate of over US$600 million [162,163]. PRRSV can affect pigs of all ages but the clinical manifestations vary, being reproductive failure in sows and respiratory difficulties and pneumonia in piglets. Mild symptoms and asymptomatic infections are common. A number of commercial vaccines are routinely used for PRRSV, some of which are administered intradermally (Table 1) [164,165]. PRRSV vaccines are mostly modified live vaccines, with some inactivated vaccines also licensed. However, most live attenuated and inactivated vaccines induce weak immune responses and are unable to cross-protect against different variants [166]. In addition, live attenuated vaccines are treated with caution due to concerns related to shedding and potential reversion to virulence [166,167]. There is a critical need to develop more effective PRRSV vaccines. The skin route of immunisation has become a focus in this effort, with promising results in eliciting stronger immune responses and enhanced protection [134,135,139] and its improved animal welfare when using needle-less technology [168]. New vaccine platforms and the use of new adjuvants and microneedle patches have been assessed for PRRSV vaccines (see below). 

Influenza virus is an orthomyxovirus. It is an enveloped, negative strand RNA virus, containing a genome segmented into eight RNA strands. Influenza viruses constantly drift and escape from pre-existing immunity and sometimes shift to create new strains to which the population is immunologically naïve. Influenza virus infection causes disease and death in humans (human influenza virus) and pigs (swine influenza virus; SIV or SIAV), among other species. It can be highly contagious and causes respiratory illness in both species. While swine usually experience mild infection and low mortality [169,170], in humans it can vary between mild to severe symptoms and mortality can be high depending on strain and susceptibility [171], with an estimated average of 650,000 deaths worldwide every year [172]. Morbidity can be high in both species, with a high disease burden observed in humans, estimated in an annual economic cost of US $11.2 billion [173]. The financial consequences of influenza in the pig industry are difficult to determine. However, the general loss of weight [174] coupled with an estimated global seroprevalence of 49.9% [175] could be an indicator of highly associated but unquantified health and financial costs of swine influenza virus infections. Despite the availability of vaccines, their effectiveness varies due to the requirement for appropriately matched vaccines to seasonally circulating influenza viruses [176]. Viral reassortment in pigs is also of major concern with respect to the evolution of viruses with pandemic potential [100,177]. Efforts are being directed towards the development of new influenza vaccines capable of eliciting a broader, more universal protective response in pigs. A skin-based approach can be beneficial and there is an increasing focus on this route for influenza vaccines in pigs. We identified 20 studies including both human and swine vaccines using the skin route [109,110,111,115,118,129,136,146,156,178,179,180,181,182,183,184,185,186,187,188] compared to approximately 60 studies evaluating either the intramuscular or subcutaneous routes for influenza vaccines in pigs, the latter also identified in a PubMed search. Some studies focus on inducing mucosal immunity by intradermal vaccine administration [115,118,182].

Pseudorabies virus, the causative agent of Aujeszky’s Disease, is the third most targeted virus, with 13 studies examining skin-based pseudorabies vaccines. This virus is an alpha herpesvirus that can affect pigs of all ages, fatal in young piglets, and presenting a wide variety of signs from respiratory disease, to reproductive failure and nervous system signs [189]. Vaccination campaigns have worked to eliminate this pathogen from domestic pigs, resulting in the majority of the literature being >15 years old. However, this pathogen still circulates in wild boars and new variants have emerged in China which escape current vaccines [190] and lead to economic losses [191]. In addition, although controversial, this pathogen can, on rare occasions, infect humans and cause encephalitis [190,192]. If dose-sparing was possible using the skin route and/or impacts on the quality of the immune response, then skin-based immunisation could potentially help tackle these escaped variants and/or a hypothetical zoonotic outbreak.

Within the “other” category we classified viruses representing less than 4% and included: porcine epidemic diarrhoea virus (PEDV), Hepatitis C virus (HCV), African swine fever virus (ASFV), Respiratory syncytial virus (RSV), Human immunodeficiency virus (HIV), rotavirus and poxvirus. 

Within targeted bacteria, the most commonly reported is *Mycoplasma hyopneumoniae* (38%). This is an important agent associated with respiratory health problems in pigs. Currently available vaccines have sub-optimal efficacy against infection, disease, and transmission. Experimental vaccines are under development [193]. Studies targeting enterotoxigenic E. coli (ETEC) represented 23.8% of all publications targeting bacterial diseases. This bacteria is still a major cause for concern for humans in low- and middle-income countries where access to preventive and therapeutic treatment is limited [194], and for farm pigs where a universal vaccine is lacking [195]. The remaining mentioned bacteria consisted of *Mycobacterium avium, Lawsonia intracellularis, Mycobacterium tuberculosis, Group E Streptococci, Chlamydia trachomatis and Actinobacillus pleuropneumoniae* (Table 2).

Only two parasites were the target of a vaccine, *Toxoplasma gondii* [196] and *Schistosoma japonicum* [197,198] both of which are zoonotic pathogens.

From this analysis, it is clear that the most reported pathogens in both the virus and bacteria category correspond to causal agents of pig diseases; PRRSV, PRV and *M. hyopneumoniae*, and to a lesser extent, pathogens that are both shared between pigs and humans, namely influenza virus and ETEC. Overall, the majority of skin-based vaccine research in pigs focuses on veterinary-specific routine immunisation in pigs, with only a minor focus on the pig as an animal model for human diseases.

**Table 2 vaccines-11-00450-t002:** Reference and year of publication of papers using skin-based vaccines in pigs according to the target pathogen.

Pathogen	References	Publication Year
Model antigens	[117,144,199,200,201,202]	2016–2021
*A. pleuropneumoniae*	[203]	2008
ASFV *	[132]	2021
*C. trachomatis*	[204]	2012
CSFV *	[49,120,128,137,145,205]	1948, 2002–2011
ETEC *	[147,150,206,207,208]	2004–2008
FMDV *	[114,130,131,140,154,209,210,211,212]	1971, 1999–2009, 2018–2020
Group E streptococci	[213]	1973
HBV *	[83,142,178,214,215,216]	2002–2003, 2009–2017
HCV *	[217,218,219]	2006, 2016, 2019
HIV *	[149]	2006
Influenza virus	[109,110,111,115,118,129,136,146,156,178,179,180,181,182,183,184,185,186,187,188]	1998–2002, 2013–2022
*L. intracellularis*	[119,220]	2020, 2021
*M. avium*	[127,221]	1983, 1978
*M. hyopneumoniae*	[51,113,133,153,160,161,220,222]	2012–2022
*M. tuberculosis*	[223]	2015
PCV2 *	[108,116,153,159,160,161,220]	2008, 2020–2022
PEDV *	[121,122]	2017, 2021
Poxvirus	[224]	1989
PRRSV *	[66,76,88,112,123,134,135,139,141,151,155,157,168,220,225,226,227,228,229,230,231,232,233]	2003–2009, 2013v2022
PRV *	[50,124,125,126,138,148,152,158,234,235,236,237,238]	1991–2000, 2005, 2011, 2016
Rotavirus	[239]	2016
RSV *	[143]	2016
*S. japonicum*	[197,198]	2000, 2010
*T. gondii*	[196]	2008

* Abbreviations: ASFV; African Swine Fever Virus, CSFV; Classical Swine Fever Virus, ETEC; Enterotoxigenic E. coli, FMDV; Food-and-Mouth Disease Virus, HBV; Hepatitis B virus, HCV; Hepatitis C virus, HIV; Human Immunodeficiency Virus, PCV2; Porcine circovirus 2, PEDV; Porcine Endemic Diarrhoea virus, PRRSV; Porcine Reproductive and Respiratory Virus, PRV; Pseudorabies virus, RSV; Respiratory Syncytial Virus.

### 3.3. Vaccine Platforms and Adjuvant Systems

We next determined the frequency of different vaccine platforms that are administered via the skin to pigs. We classified vaccine platforms as live attenuated (weakened organisms), inactivated (killed organisms), nucleic acid (DNA or RNA), subunit (including subunit, virus like particles (VLP) and toxoid vaccines) and virus vector vaccines. 

The frequency of four of the five main platforms in skin-based immunisation in pigs is practically equivalent within the reviewed published papers (Figure 4). 

Live attenuated and inactivated platforms have been the traditional vaccines in veterinary medicine. However, advances in the field have led to the development of new platforms that are increasingly being used for pig viral vaccines [240]. DNA vaccines, which consist of plasmids encoding the antigen of interest, present advantages such as faster re-derivation of the vaccine to new strains and the inability to revert to virulence. The capacity to use specific antigens instead of the whole organism also permits the differentiation of infected from vaccinated animals (DIVA) [241,242]. Moreover, DNA vaccines can overcome maternal antibodies in the neonate pigs via the skin-based route [115,182]. Some approaches identified in the literature utilise DNA-based vaccines with the aim of improving heterologous protection to influenza [109,115,182]. However, only one DNA vaccine has been licensed for equine use, and none for pigs. This is largely due to a limited immune response observed in large animal models [241,243]. Nevertheless, advances in the field, including better vectors, new delivery methods, and use of adjuvants, have led to improved humoral and cellular immune responses to vaccination observed in pigs and in clinical trials [244,245]. The first DNA vaccine, for SARS-CoV-2, received approval for human use in 2021 in India [246,247]. This vaccine is administered into human skin. 

The European Medicines Agency (EMA) defines adjuvants as ingredients in a medicine that increase or modify the activity of the other ingredients. Adjuvants are used to enhance and modulate vaccine-induced immunity. Over 60% of identified papers using skin-based vaccination in pigs included adjuvants; the majority mixed or co-inoculated with the vaccine [51,76,88,108,110,112,113,115,117,118,119,120,121,122,123,124,125,129,131,132,133,134,137,138,139,140,141,142,143,144,146,151,153,154,156,158,159,160,161,168,179,180,182,186,187,188,196,198,199,201,202,203,206,208,210,211,212,213,214,215,218,220,221,222,226,227,228,229,232,233,236] and on a few occasions administered separately [147,178,206,207], to modulate the immune response. Numerous studies formulate multiple adjuvants together [51,108,113,117,119,133,139,146,156,182,202,220,222,226,227] while others compare safety and/or efficacy of different adjuvants [88,120,121,132,144,146,156,180,186,198,199,201,206,211,213,220,227,228]. We examined the proportional use of different adjuvants by the skin route in pigs. We classified them based on the EMA definitions [248] (Figure 5). Emulsions include surfactants, oils, oil-in-water, water-in-oil, water-in-oil-in-water and oil-based emulsions. Endogenous immuno-modulators include cytokines, CTLA4, perforin and cGAMP; and TLR agonists constitute a separate category. Microbial derivatives consist of cholera toxin, enterotoxin and cdAMP; particulate adjuvants include liposomes and nanoparticles. A final category includes alum-based adjuvants. The adjuvants not fitting any of these categories were classified as “other”, including vitamin C, polymers and phosphazene. 

The most commonly reported adjuvants are emulsions (49.4%, Figure 5). Emulsions are formed by two immiscible liquids brought together and are a common choice of adjuvant in veterinary vaccines [249,250]. They are particularly used for livestock vaccines given their low cost, ease of preparation and use, and efficacy [250,251,252,253]. Special attention needs to be taken when choosing the type of emulsion, as the benefit of protection against virus challenge must be greater than skin-based reactogenicity, as seen for a PEDV vaccine [121] but not for other vaccines [132,253]. Some emulsion-based adjuvants have also shown to be safe for skin administration, inducing a strong response mediated by skin DCs [125,199,210]. Emulsions, like the licensed Emulsigen, have shown a more balanced local reaction and immune system activation when delivered to skin in pigs [199,254]. Various emulsion adjuvants are licensed for use in pigs, including various under the band Montanide [250,255]. There are fewer licensed human vaccines using emulsion adjuvants, likely due to unwanted reactogenicity profiles of early emulsion adjuvants used in clinical trials. Such issues have been resolved through improved manufacturing processes and better understanding of how emulsion properties are linked to reactogenicity and immunogenicity [256,257]. TLR agonists represent 12.9% and endogenous immunomodulators 11.8% of the reported adjuvants in skin-based vaccines in pigs. Substantial research has focused on developing adjuvants based on innate sensing of pathogen patterns, such as the TLR, STING and NOD systems [88,187,228,258]. Alum was the first adjuvant to be approved for human use, and it is widely used in humans [259]. However, it only represents a 4.3% of adjuvants reported in skin-based vaccines in pigs within the identified studies. A possible explanation could be the potential for causing granuloma if incorrectly administered into the intradermal space and/or its strong local reactogenicity. Contrary to what we observe in human vaccines, where adjuvants are generally reserved for subunit vaccines, many of the pig skin-based whole vaccines (live attenuated and inactivated) incorporate adjuvants. An example of this is the commercial vaccine Porcilis PRRSV consisting of a modified live virus (MLV) accompanied by dl-α-tocopherol acetate adjuvant mixed to form an emulsion [76,141,151,168,220,232,233]. 

Therefore, similar to human vaccines, new adjuvants suitable for skin use are being developed, however, a number of adjuvanted vaccines, particularly emulsions are routinely and safely used in pigs.

### 3.4. Routes and Devices 

There are various skin administration routes that have been defined by regulatory bodies such as the FDA. The epidermal and intradermal route delivers vaccines into the epidermis and dermis, respectively. Transdermal routes administer material through the dermal layer of the skin to systemic circulation by diffusion. Percutaneous is defined as delivery through the skin, whereas epicutaneous is delivery onto the surface of the skin [260]. 

Intradermal immunisation accounts for 82.6% of the routes used in the included studies (Figure 6). While there is no specific platform that stands out for the intradermal route, 83% of epidermal vaccinations use DNA platforms. Some of the studies evaluated the combination of the skin-based route of administration with intramuscular, oral, subcutaneous, or intranasal, simultaneously [112,132] or through heterologous prime-boost regimens using the same or different vaccines [111,122,142,147,150,202,206,208,217,223,229,234,238,261], with the objective to elicit a stronger immune response. 

A range of delivery technologies are used for skin-based administration in addition to conventional needle-and-syringe (Figure 7). Jet injectors were the most frequently used device (35%). These are needle-free devices used to deliver liquid vaccines or drugs into intradermal, subcutaneous, or intramuscular tissues by using high pressure to create a narrow stream that can penetrate the skin. They have been used since the 1950s [262,263], although the use of the first generation of injectors was stopped due to rising concerns of iatrogenic transmission of pathogens between individuals. These issues were overcome in the newer generation of jet injectors [264] which present various advantages, especially in the pork industry. These advantages are mainly shared with all needle-free devices, compared with the classical needle and syringe, including ease of administration, no generation of hazardous sharps waste, elimination of needle-related accidents, reduced pain and stress, and better reproducibility [108,265,266]. The MSD IntraDermal Application of Liquids (IDAL) device was used in 26 of the “skin immunisation” studies reviewed; accounting for 56.5% of the jet injectors mentioned. IDAL is an example of a needle-free jet injector that has been in use since the 2000s [78,79,267]. It is approved for administration of vaccines against PCV2, PRRSV and *Mycoplasma hyopneumoniae*, some of the most common infections in pigs [78,267] (Table 1), and has proven to be beneficial in animal welfare [108]. Jet injectors have been demonstrated to result in a less painful and less aversive experience compared to intramuscular injection, proven to further enhance animal welfare [108,225]. By controlling several factors, including pressure applied, size of orifice, angle of injection and knowing the thickness of the skin, jet injectors can target the dermis [268] and, thus, are used for intradermal vaccination. Intradermal administration was used with jet injectors in the majority of studies reviewed.

Needle and syringe, including short intradermal needles, represent 17.9% of the devices mentioned. Microneedle(s), both in array and single format, constitute 10.7% of the devices. Skin structural changes, by means of skin ablation or electroporation, is reported with a frequency of 7.1% and usually accompanies other devices, such as jet injectors, needles, or microneedles (Figure 7). 

Gene guns, including ballistic injectors, were assessed at a frequency of 7.9% (Figure 7). Most immunisations use the epidermal route use a gene gun, with the exception of one study using a combined laser and array patch [201] and one comparing gene gun to electroporation [109]. Gene guns propel DNA-coated gold particles into the skin using gas pressure at 400–600 psi (154). They are limited to targeting the epidermis [66,269]. Finally, the category “other” includes all devices/techniques used in less than 5% of cases. Condensation chambers, scarification and cover slip methods conform this category, and correspond to the epicutaneous (the former) and percutaneous (the other two) routes of administration [126,143,197,224].

A large proportion (18.6%) of publications did not state the device that was used or the method of skin administration and are categorised as “missing”. The only detail mentioned is the use of an intradermal route. This lack of detail and the use of a potentially generic term of “intradermal” could bias the higher frequency of studies using the intradermal route as opposed to the other skin-based administration routes (Figure 6). The lack of specific details prevents the re-use and reproducibility of these findings. 

### 3.5. Induction of Immunity

#### 3.5.1. Adaptive Immune Responses

Out of the 117 papers, 10 focus on safety aspects with no mention of the immune response [108,130,149,154,160,168,185,200,201,216] and 3 focus on the innate immune response [88,199,207]. The remaining 104 papers assess vaccine-induced adaptive immunity and/or efficacy. 

#### 3.5.2. Humoral and Cellular Immune Responses Elicited by Skin-Based Immunisation

Systemic immune responses

Antigen-specific antibody responses were the most frequently evaluated adaptive immune response; only 6 out of 104 publications did not analyse humoral immunity. Systemic antibody responses are predominantly assessed, including antigen and/or pathogen-specific antibodies, neutralizing antibodies, and hemagglutinin inhibition (HI) titters for influenza virus. T cell responses were evaluated in less than half of the studies while only eight publications investigated B cell responses [50,122,133,204,206,208,219,227]. 

Most publications evaluated PRRSV-specific antibodies (20 out of 23 papers) and T cells (15 out of 23 papers), whereas only one paper evaluated B cell responses. Out of the 20 studies that examined PRRSV-specific antibodies subsequent to skin-based immunisations, 12 reported achieving seroconversion after vaccination [76,134,135,139,141,151,155,157,220,225,226,233]. Seroconversion is defined as sample to positive (S/P) ratios above a 0.4 threshold. From those 12 studies, 11 studies assessed a live attenuated vaccine and 1 tested a DNA vaccine. Strong antibody responses, especially after challenge, were induced in a number of studies, for example, after three DNA immunisations [66] or subsequent to two adenoviral vector-based vaccine immunisations [231]. Compared with placebo, strong humoral responses induced by skin-based immunisation led to better protection after homologous, heterologous virus strain challenge or natural infection, shown as a decrease in clinical signs, lung lesions, viral shedding or viremia [76,112,134,135,139,157,220]. However, increased antibodies do not always correlate with protection, as observed by partial, limited or no efficacy of a subunit [123] or an inactivated vaccine [227] after challenge despite an increase in specific-antibodies. Furthermore, some PRRSV vaccines, such as an adjuvanted inactivated vaccine delivered into the skin, did not induce seroconversion even after challenge, and no protection was observed [228]. Protection, however, is not necessarily only linked to antibodies, as cellular immune responses to PRRSV play an important role as well [76,112,135,139,141,157]. Seroconversion and efficacy with no significant differences in protection against challenge have been observed between the single vaccines and multivalent vaccines, the latter of which can contain PCV2, *M. Hyopneumoniae* and *L. intracellularis* [220]. 

The second most frequently assessed vaccine indication in the literature is influenza virus. Studies that examined skin-based delivery of influenza vaccines (n = 20) focused mainly on evaluating the antibody response after vaccination (n = 19); seven studies evaluated T cell responses, and none evaluated B cell responses. A virus challenge was performed in 11 of these studies. For this pathogen, antigen-specific antibodies; mostly IgG, neutralizing antibodies, and hemagglutinin inhibition (HI) titers were commonly evaluated. A little over half of the studies (n = 11) on influenza used DNA vaccines for immunisation [109,110,111,115,118,136,146,181,182,184,188], with the remaining studies testing inactivated [129,178,179,183,186] and subunit vaccines [156,180,187]. Independently of the platform used, large increases in virus-specific antibodies after skin-based vaccination were reported in 15 studies, with 12 of them also reporting HI antibodies’ greater than the seroprotective threshold and/or neutralizing titters [109,110,115,118,129,136,156,178,180,181,182,183,184,186,188]. Significant increases in IFNg-, IL17- or IL13-secreting cells or proliferation recall responses in blood or lymph nodes in vaccinated compared to controls were observed in 3 studies [110,129,186], while no T cell responses were observed in response to a subunit vaccine [156]. A study with an inactivated AS03-adjuvanted vaccine showed increase in antibodies but considerable variability observed between individual animals [179]. Two studies evaluating a DNA [146] or a subunit [187] vaccine adjuvanted by microbial derivatives or endogenous immunomodulators, point towards increased antibody titters compared to unadjuvanted vaccine or HI titters higher than the threshold, respectively. Antibody responses are mostly associated with enhanced protection to both homologous and heterologous virus challenges, described as reduction or elimination in viral shedding, load or clinical signs [109,115,136,182,184,186,188]. Nevertheless, a subunit vaccine, using conserved influenza antigens fused to an anti-CD11c antibody adjuvanted with CpG resulted in exacerbation of disease after challenge despite a significant increase in antibodies after intradermal administration of the vaccine compared to controls [180]. Failure to protect pigs either against homologous or heterologous virus challenges, despite higher specific-antibody titters induced after immunisation with inactivated and DNA vaccines [118,129], points towards other type of responses needed to induce protection, namely cellular responses. 

Similar to PRRSV and influenza, all 13 studies targeting pseudorabies virus (PRV) focused on antibody responses, whereas only 5 evaluated T cells [50,124,125,152,235] and one assessed B cell responses [50]. Neutralizing antibodies were the main response evaluated after PRV vaccination. Seroconversion according to neutralizing titters after skin vaccination with DNA, subunit, live attenuated or viral vector vaccines was observed in 9 trials, with differences being observed between pre- and post-vaccination stage and/or between vaccinees and controls [50,124,125,126,138,148,152,235,236]. Antibodies increased after challenge. Significant increase in systemic antigen-specific IFNg-secreting T cells or lymphoproliferation responses compared to controls was observed in 2 out 5 studies evaluating T cell responses [125,235], with the other three showing increases but high variability or insufficient statistical power [50,124,152]. Positive correlation between neutralizing antibody titters and T cell responses was demonstrated in one study [125]. At least partial protection, and/or reduced virus shedding was often seen after vaccination and challenge [50,124,126,138,148,152,235]. However, lack of correlation of protection between neutralizing titters and/or cellular responses and protection has also been observed [124,236], underlining the importance of other responses. Additionally, a significant increase in IgG1- and IgG2-secreting B cells in skin-draining lymph nodes, after immunisation with an unadjuvanted subunit vaccine, has been demonstrated by ELISPOT [50].

Vaccine-induced B cell responses at a systemic level tend to differ between studies. Two studies demonstrated a significant increase in IgG-and IgA-secreting B cells in the blood and draining lymph nodes of animals intradermally vaccinated with subunit and inactivated vaccines when compared to controls. This B cell response additionally, correlated with protection [50,133]. Memory B cells have also been detected in the blood of pigs vaccinated with a subunit HCV vaccine [219]. However, lack of significant increase in percentage of B cell [227] or low correlation with protection [204] has also been reported.

Mucosal immune responses

It has become increasingly evident that mucosal immunity can be required for sterile protection against infection. Intradermal vaccination has shown potential for inducing this kind of immunity in various species, including humans [270]. Within the papers analysed in this review, 14 studies have documented vaccine-induced mucosal immunity. These mucosal samples include saliva [49], bronchoalveolar lavage (BAL) [51,115,182], nasal mucosa [50,118,202,204,226,229], colostrum [121], gut [122,206,208] and vaginal mucosa [202,204]. 

Very few studies have looked at mucosal antibodies after PRRSV vaccination. Although a certain tendency to higher nasal IgG was hinted after skin-based immunisation with a DNA vaccine alone, or in combination with a live attenuated vaccine by the intramuscular route, levels remain low and not significant, with no IgA detected [226,229]. A slight increase was also reported for nasal and BAL antibodies (IgG and IgA) after influenza vaccination with DNA vaccines and challenge [115,118,182]. In the case of vaccination against PRV, a significant increase in nasal IgA after vaccination and intranasal challenge compared to controls was demonstrated using a subunit vaccine platform. Specifically, a positive correlation between protection and amount of nasal IgA was observed, although to a lesser extend than that observed for serum IgG [50]. Furthermore, inactivated and viral vector vaccines targeting *M. hyopneumoniae*, Classical Swine Fever virus and Porcine Epidemic Diarrhea Virus have been able to elicit mucosal antibodies in pigs after intradermal immunisation by needles or jet injector [49,51,121]. Mucosal antibodies were proposed to improve survival of vaccinated sows [49] and partial protection of their piglets through colostrum [121]. 

Even less frequent is the analysis of B cell responses in mucosal tissues. Generally low levels of antibody-secreting B cells in the gut have been described [122,206,208]. A heterologous prime-boost regimen consisting of DNA intradermal priming with subunit oral booster reported the induction of antibody secreting cells (ASC) in various mucosal lymphoid tissues [206,208]. Therefore, between the 8 studies evaluating B cell responses, the most documented significant effects on these cells are observed in the blood or administration site draining lymph nodes, while only weaker or no responses are identified in mucosal tissues. 

Further studies addressing mucosal immunity and B cell responses induced by skin vaccination are needed to better understand the magnitude and mechanisms of action of this immunisation route.

### 3.6. Comparison of Routes of Immunisation

Many of the published papers administering vaccines through the skin route in pigs compare this route to the more predominant intramuscular route, with fewer articles comparing to intranasal and subcutaneous administrations. 

Similar humoral and/or cellular responses and/or protection are achieved either by intradermal or intramuscular vaccination against PRRSV using live attenuated platform with or without adjuvants, with intradermal being superior in some cases [76,134,135,139,141,151,155,225]. These studies all used live attenuated PRRSV vaccine. However, on two occasions, the intramuscular route was shown to be more immunogenic compared with the skin route with inactivated vaccines [227,228]. Similarly, in PRV skin-based immunisation with DNA, live attenuated, viral vector and/or subunit vaccines, comparable or even stronger and faster immune responses were observed with skin-based immunisation [50,125,126,138,152,158,235,236,236]. An adjuvant sparing effect was observed in one trial with a subunit vaccine administered by the skin route [50]. Influenza subunit [180], inactivated [179] or DNA vaccines [136], however, induced higher responses when the intramuscular route was used, including the prevention of disease exacerbation. On occasion, skin-based immunisation showed encouraging results [186]. 

Heterologous prime-boost administration strategies featuring skin immunisation with other routes or the use of different vaccines in the prime and boost have been tested in the pig model. An increase in antibody or cellular responses coupled with clinical protection was observed in a heterologous intradermal-intramuscular strategy in PRRSV [229,230], influenza [111] and PRV [234] vaccination, when compared with each route on their own or in combination with intranasal vaccination. The simultaneous use of these routes with adjuvanted inactivated or DNA vaccines seems to also benefit the induction of antibody responses [112,238]. Epidermal and intradermal skin administration for DNA vaccines has also shown superiority in terms of antibody production and cellular responses, compared to the combination of intradermal and intramuscular routes in PRRSV [66] and PRV [235] trials. 

Overall, there is some evidence of skin-based immunisation being similar or superior to the classical intramuscular route. A clear advantage in reduced intradermal doses has been observed, reaching comparable responses to those observed in intramuscular immunisation and even surpassing those induced by higher intradermal doses [203]. 

However, despite the encouraging evidence favouring skin-based immunisation, it will most likely take time for new routes to replace intramuscular administration, which continues to be the most frequently licensed route in the pig industry (Table 1). Practically, immunization of pigs should be efficient, easy to perform on each animal and across a herd, and it should not decrease the animal’s welfare or health. Ideally, it would induce higher quality, durable responses, potentially with lower vaccine doses. Conventional intradermal vaccination by needle-and-syringe is time consuming and prone to variability. The new needle-free technologies can be more expensive than use of a needle-and-syringe by the intramuscular route. This underscores the need for further development and optimisation of vaccine delivery technologies to the skin route to enhance vaccine-induced protection against pathogen challenge.

### 3.7. Safety and Adverse Events

Safety is a crucial parameter to consider both for the pig industry and possible translation into human medicine. Skin-based immunisation has been associated with local and sometimes general adverse events [271]. From the 117 papers included in this review, 63 evaluated safeness associated either to the vaccine platform, adjuvant or device used in skin-based immunisation. 

The general systemic side effects of vaccination, including fever, weight loss and fatigue, were only mild or absent in the pigs vaccinated using the skin route [110,113,123,131,132,134,135,138,140,151,153,157,158,160,179,188,203,221,222,227,228,233,239]. In addition, co-administration of multiple vaccines into the skin has shown the same safety profile as individual vaccines for PCV2, PRRSV, *M. hyopneumonia* and *L. intracellularis* [153,220], which is not only convenient for the farmer but can also reduce the stress of handling and potential pain to the animals. Special attention is given to the safety of newly developed vaccines for skin administration. Live attenuated viral strains of PRV [234,238], PRRSV [76,139,155] and FMDV [130] were shown to be safe, as observed by the absence or mild presence of clinical signs, lack of transmission, no recovery from tissues and limited dissemination of vaccine virus, when administered intradermally. Moreover, studies with poxviruses in the 1980s and 1990s [126,224] evaluated their possible use as viral vector in the pig, proving their safety. A virus replicon particle for CSFV [145] has proven to be a safe candidate for vaccine development after showing mild or no clinical signs and inefficient replication after skin administration. In addition, DNA vaccines for FMDV [131], HBV [216] and PRRSV [157] are also well tolerated by the skin route, with acceptable biodistribution and no toxicity. 

Local reactogenicity after intradermal vaccination generally includes swelling, redness, erythema, induration and skin lesion. Skin-based vaccine administration is more frequently associated with local reactions than the intramuscular or subcutaneous routes, as reactions in skin can be observed compared to reactions that occur deeper in the body such as the muscle [132,140,179,180,203,214,215,222,224,227,228]. Unwanted reactogenicity could be decreased by the use of smaller volumes of vaccine [210]. However, local side effects are particularly detected when needle and syringe or scarification techniques are used. Thus, the development of alternative devices for skin immunisation is important to guarantee decreased reactogenicity. For instance, needle-free jet injectors have proven to reduce local side effects and avoided pathogen transmission between animals [155], leading to better tolerability of the skin route. Other devices like microneedles, microneedle patches, gene guns or electroporators can also prevent severe skin lesions or any prolonged or permanent damage [142,183,216,228].

An appropriate balance is needed between immunopotentiation and safety when choosing adjuvants for skin immunisation. Acceptable transient skin reactions or oedema are commonly observed after skin vaccination with diverse type of adjuvants, including liposomes, emulsions, polymers, endogenous immunomodulators and microbial derivatives [117,121,132,187,210,214]. Strong reactogenicity has been associated with alum-based adjuvants and certain emulsions such as highly reactogenic Freund’s adjuvants [132,199]. TLR7/8 agonists, such as resiquimod, induce macroscopic and microscopic lesions that are milder or absent for TLR1/2 or TLR9 agonists (Pam3Cys or CpG oligodeoxynucleotides) [88,228]. Unwanted local reactions can be mitigated by adjusting the amount of adjuvant used [186], or adapting to different skin administration methods such as topical application or epidermal powder delivery [178,201]. However, this must be balanced with maintaining or improving potency and efficacy.

Overall, the skin route of vaccination in pigs is generally safe, with mild or no systemic adverse events recorded. However, balance needs to be achieved to improve animal’s welfare, industry benefits and translatability into human medicine, when relevant. 

## 4. Future and Directions for Pigs in Skin Immunisation

The skin represents an attractive target for immunisation. Given the similarity between pigs and humans coupled with the central role of pigs in the One Health approach, the pig represents a very relevant animal to examine vaccine-induced immunity by skin immunisation. Within our PubMed search, a predominance of veterinary vaccine research, specifically focusing on viral targets, has been noted as being opposed to pigs being used as a model for human vaccines. This could be explained due to an increased interest in this route for its potential to facilitate immunisation and increase the welfare of farm pigs. These veterinary vaccine findings emphasise the advantages of skin immunisation for humans. In particular, the safe use of adjuvants, particularly emulsion-based adjuvants in pigs, could set a precedence for skin-based vaccines for humans. Further emphasis should be put on the pig as a model for skin immunisation, as given its common characteristics with humans and its position as a source of zoonotic diseases, the potential for translatability increases. 

Multiple routes of administration into the skin exist, but a lack of precision in reporting and clarity in devices used makes it difficult to evaluate potential differences in elicited immune responses. Better adherence to the official definitions of routes and further details when describing immunisation could help further characterise immune response after skin immunisation and improve the reproducibility and re-use of published results. 

Despite the promising immune responses observed, future research should focus on the underlying mechanisms of action and anatomical breadth of responses to further understand how vaccines administered to the skin are working in systemic and mucosal compartments. In addition, a better understanding of how immune responses correlate with protection is needed. The development of a wider range of reagents specific for pig assays could help overcome challenges in this field. 

Despite the predominance of more classical routes of immunisation, increased interest in the skin within the pig industry coupled with the further development of administration technologies should lead to the increased use of skin-based immunisation in the future. Lessons learned from these farm animals can also benefit human medicine. 

## Figures and Tables

**Figure 1 vaccines-11-00450-f001:**
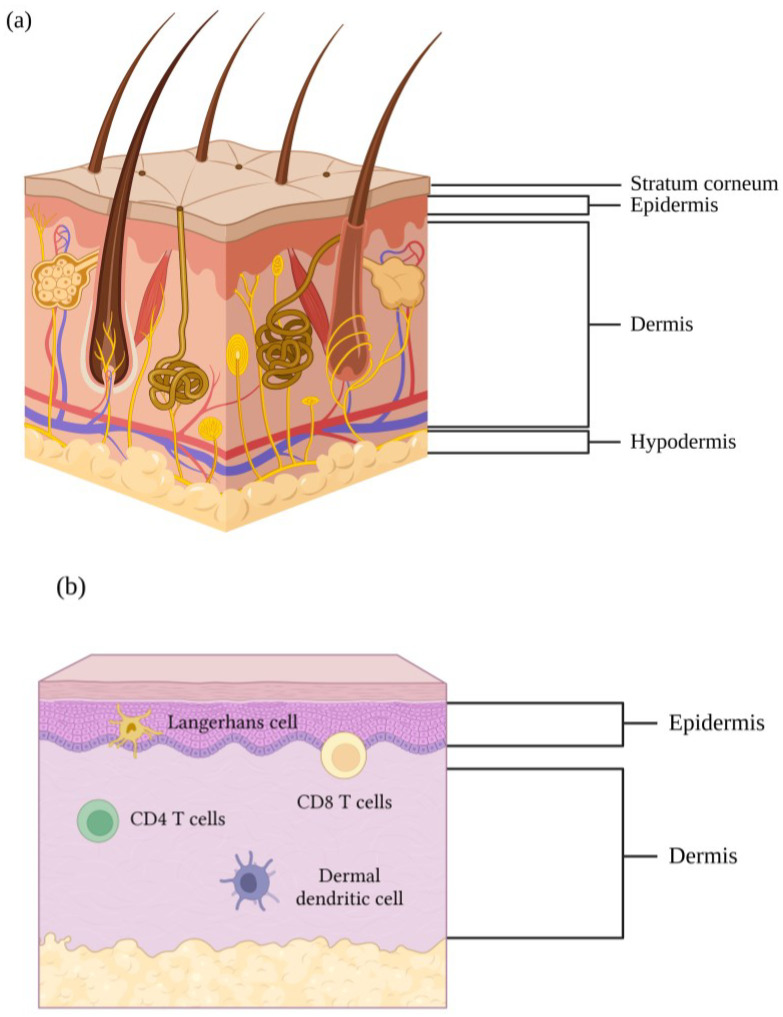
(**a**) Schematic representation of the structure of the skin. (**b**) Schematic representation of immune cells location in the skin. Created with Biorender.com.

**Figure 2 vaccines-11-00450-f002:**
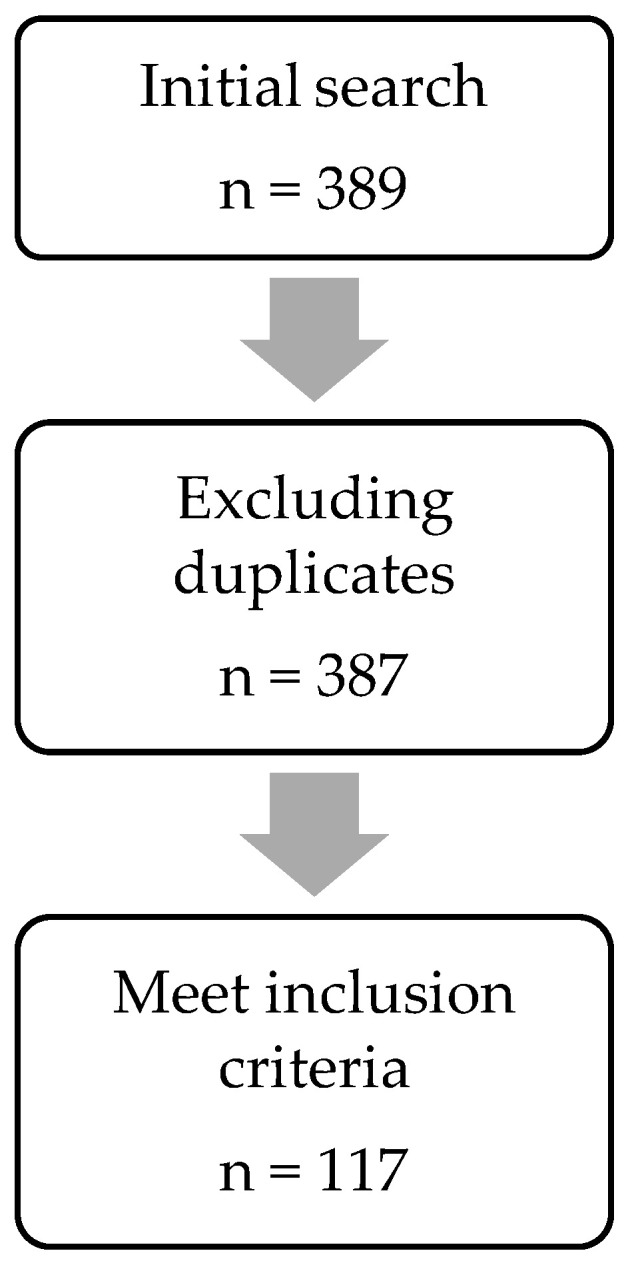
Number of papers identified, and numbers excluded during analysis.

**Figure 3 vaccines-11-00450-f003:**
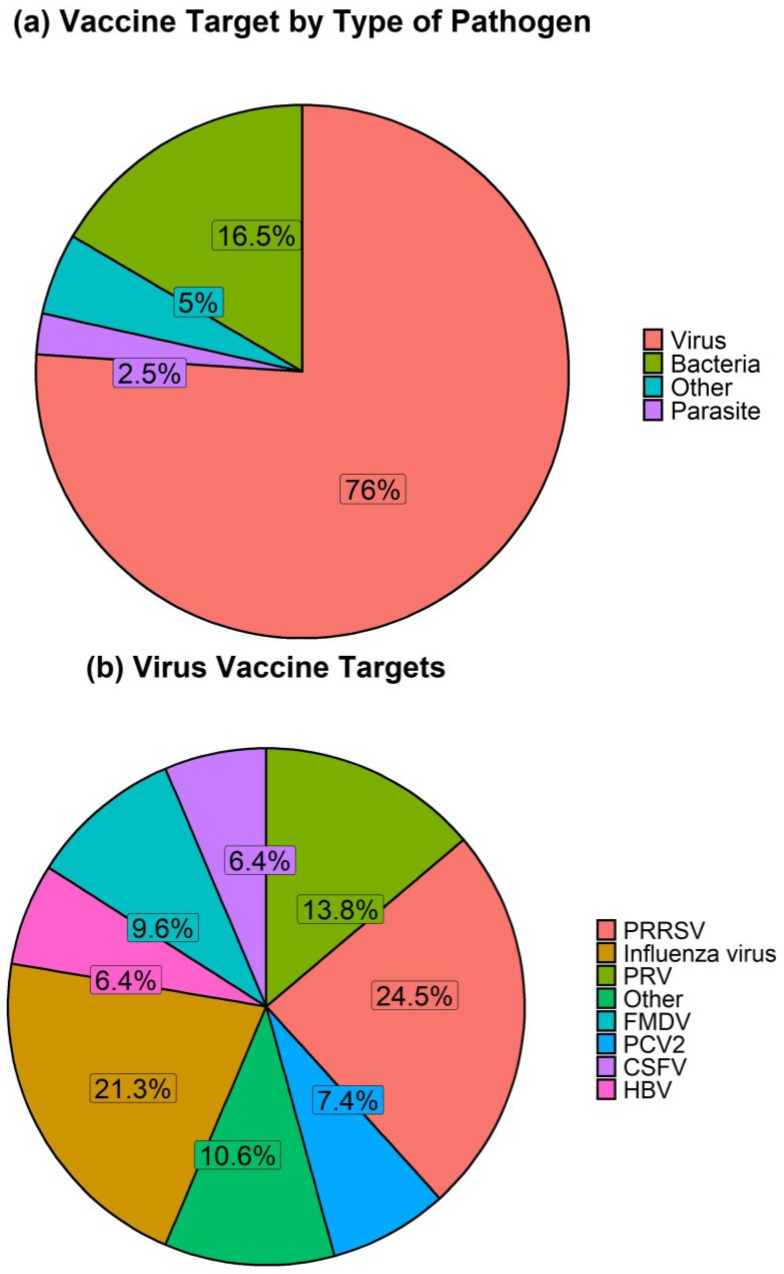
(**a**) Frequency of skin-based vaccination used in pigs according to whether the target of the vaccine was a virus, bacteria, parasite or other pathogen, as identified in 117 published papers. (**b**). Within viruses, frequency of skin-based vaccination used in pigs according to disease indication in 92 published papers. PRRSV: Porcine Reproductive and Respiratory Syndrome Virus, PRV: Pseudorabies virus, FMDV: Food and Mouth Disease Virus, HBV: Hepatitis B virus, CSFV: Classical Swine Fever Virus, PCV2: Porcine Circovirus 2.

**Figure 4 vaccines-11-00450-f004:**
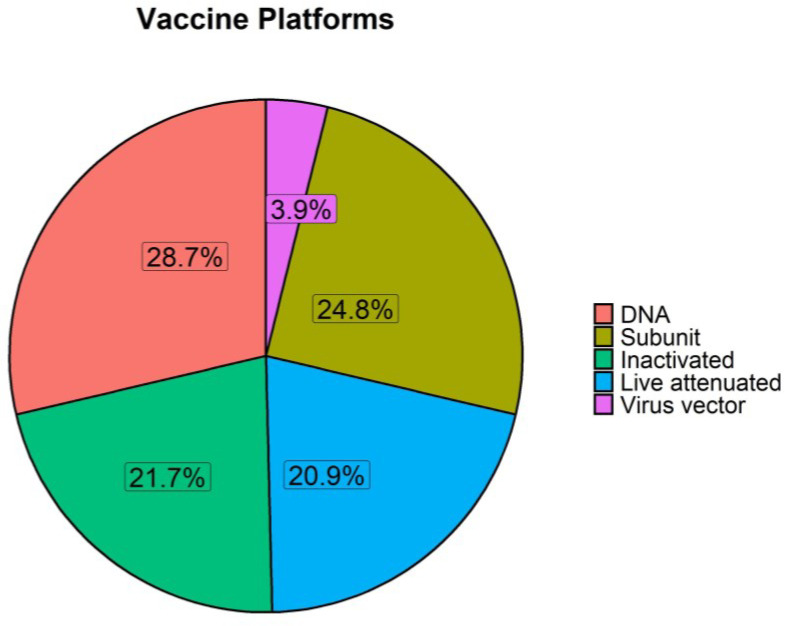
Frequency of use in 117 published papers of skin-based vaccine in pigs according to the type of vaccine platform.

**Figure 5 vaccines-11-00450-f005:**
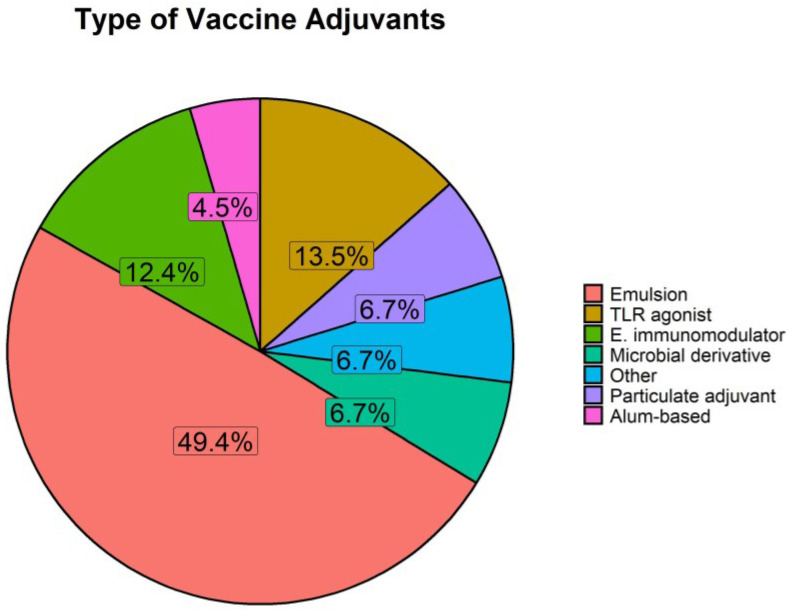
Frequency of use of adjuvant types in skin-based vaccines in pigs in 117 reviewed papers. E. immunomodulators; endogenous immunomodulators.

**Figure 6 vaccines-11-00450-f006:**
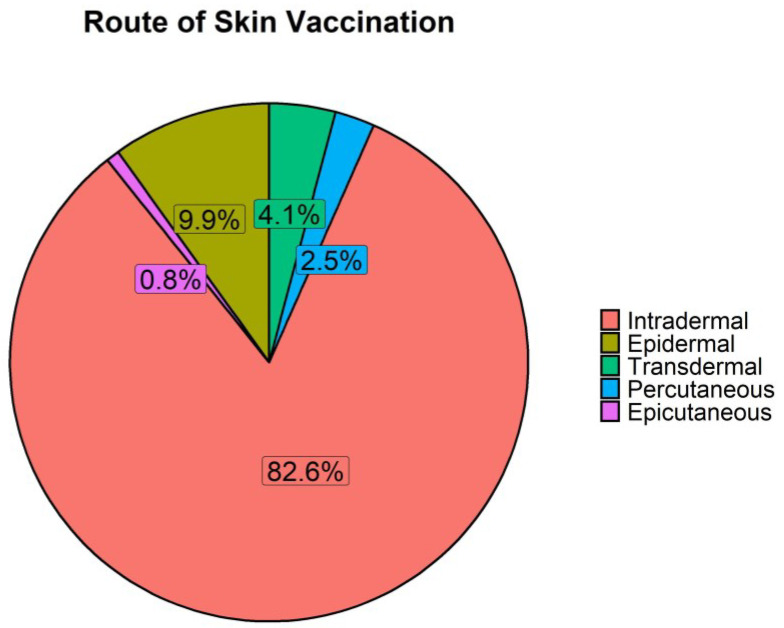
The proportion of different reported skin-based routes of administration, or based on the device specifications (pressure, needle length) in the 117 published papers reviewed.

**Figure 7 vaccines-11-00450-f007:**
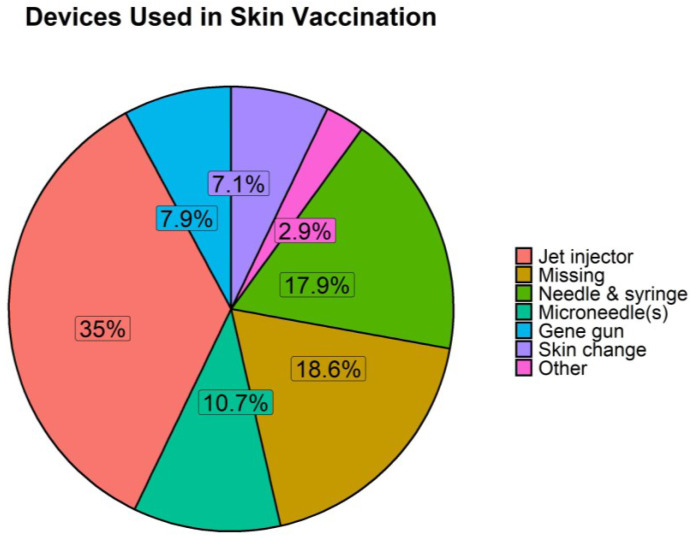
The proportion of different reported devices used in skin-based route of administration in the 117 published papers reviewed.

## Data Availability

Not applicable.

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
