# Peer review of "Skin-Based Vaccination: A Systematic Mapping Review of the Types of Vaccines and Methods Used and Immunity and Protection Elicited in Pigs"

_vaccines, 2023, doi:10.3390/vaccines11020450_

Round 1
Reviewer 1 Report
For the Figure 1 it is possible major quality? In th eFigure 2 top of the box is missing. For the Figure 3 it is necessary to specify better in the text.
In the paragraph 1.2. the number of references have different carcater.
There are missing spaces between words (line 165, 604), extra parentheses (line 208), ortograpfhy (PRRRSV at line 515),
Author Response
Attached as PDF file.

Reviewer 2 Report
This paper consists of a scientific review that focuses on the use of the intra or transcutaneous application of vaccines for swine species and its effect on innate and immune response. The topic is of great interest for veterinary and general vaccinologists as well. There is not much published on this topic and the extensive mapping of citations and articles contributed by this review constitutes a timely and useful tool for all those people interested in advancing knowledge on the subject. While the review is written with the biomedical and One Health approach points of view in mind, the most significant contribution of this paper is in the area of veterinary (swine) vaccines. The general introduction to the skin as a site favoring innate and acquired immunizations is appropriate. The number of references is exhaustively finished, having included a very comprehensive list of articles involving ID inoculation of immunogens , their adjuvants and the strategies or platforms used in each case. It should be noted however, that the application of ID vaccines has advanced significanly more in veterinary medicine rather than in human medicine. Unfortunately, those few papers that actually compare the ID route of administration of immunogens in pigs with other more conventional routes such as IM have not been critically appreciated or reviewed by the authors of this review. It is then left up to the readers to go to those few cited papers that indeed compare routes of immunization in pigs to get their own conclusions. Fortunately, the extensive character of the listing of pertinent references contained in this review reveal the existence of those articles to the readers. The English language and terminology used in the paper is quite acceptable. I would only recommend a single but important correction in one of the most significant parts of this review: the abstract. In line 10/11 of page 1 of the paper: I suggest replacing the term "veterinary animals" by the more correct " veterinary medicine" or "veterinary species"
Author Response
Attached as PDF file.

Reviewer 3 Report
This is a systematic review where the authors have discussed types of skin based vaccines of pig with special emphasis on vaccines, methods and immunity and protection etc. It’s a very well-described and comprehensive review and the sections are well-organized.
My comments are as follows:
Please add a figure showing the Dendritic and Langerhans cells in skin histology
METHOLOGY:
Please mention the review strategies. What strategies did you follow during your systematic review?
As it is a systematic review, you should show the steps of PRISMA guidelines in your figure, at least the following steps: (1) article identification by initial search and removing duplicates, (2) Article screening based on title and abstract, (3) Article eligibility according to your inclusion and exclusion criteria, and (4) Articles included.
Do the author have a registration statement for this systematic review?
Table 2, add another column and show the study period (though these are available under ref. section at the end)
Section3.4; line 424 to 428, please make clear it more clear what is the difference between the epidermal / intradermal route AND epicutaneous route. All are injecting into skin??
ALMOST NOTHING is mentioned about the disadvantages of skin based vaccination in pig? please write a para in this regard.
"3. Results" should be "3. Results and Discussion," as you showed the results and discussed them combinedly in this section.
"4.1. Future and directions for pigs in skin immunisation" under "4. Discussion and conclusions"; please clarify it.
Author Response
Attached as PDF file.
